# The Association between Ankle–Brachial Index/Pulse Wave Velocity and Cerebral Large and Small Vessel Diseases in Stroke Patients

**DOI:** 10.3390/diagnostics13081455

**Published:** 2023-04-18

**Authors:** Yu-Ming Chang, Tsung-Lin Lee, Hui-Chen Su, Chung-Yao Chien, Tien-Yu Lin, Sheng-Hsiang Lin, Chih-Hung Chen, Pi-Shan Sung

**Affiliations:** 1Department of Neurology, National Cheng Kung University Hospital, College of Medicine, National Cheng Kung University, Tainan 704, Taiwan; cornworldmirror@hotmail.com (Y.-M.C.);; 2Institute of Clinical Medicine, College of Medicine, National Cheng Kung University, Tainan 704, Taiwan; 3Department of Public Health, College of Medicine, National Cheng Kung University, Tainan 704, Taiwan; 4Biostatistics Consulting Center, National Cheng Kung University Hospital, College of Medicine, National Cheng Kung University, Tainan 704, Taiwan

**Keywords:** ankle–brachial index, brachial–ankle pulse wave velocity, large artery atherosclerosis, cerebral small vessel disease, acute stroke

## Abstract

(1) Background: The study investigated whether the ankle–brachial index (ABI) and pulse wave velocity (baPWV) could reflect the severity of small vessel disease (SVD) and large artery atherosclerosis (LAA). (2) Methods: A total of 956 consecutive patients diagnosed with ischemic stroke were prospectively enrolled from July 2016 to December 2017. SVD severity and LAA stenosis grades were evaluated via magnetic resonance imaging and carotid duplex ultrasonography. Correlation coefficients were calculated between the ABI/baPWV and measurement values. Multinomial logistic regression analysis was performed to determine predictive potential. (3) Results: Among the 820 patients included in the final analysis, the stenosis grade of extracranial and intracranial vessels was inversely correlated with the ABI (*p* < 0.001, respectively) and positively correlated with the baPWV (*p* < 0.001 and *p* = 0.004, respectively). Abnormal ABI, not baPWV, independently predicted the presence of moderate (adjusted odds ratio, aOR: 2.18, 95% CI: 1.31–3.63) to severe (aOR: 5.59, 95% CI: 2.21–14.13) extracranial vessel stenosis and intracranial vessel stenosis (aOR: 1.89, 95% CI: 1.15–3.11). Neither the ABI nor baPWV was independently associated with SVD severity. (4) Conclusions: ABI is better than baPWV in screening for and identifying the existence of cerebral large vessel disease, but neither test is a good predictor of cerebral SVD severity.

## 1. Introduction

The ankle–brachial index (ABI) and pulse wave velocity (PWV) are widely used noninvasive modalities to evaluate atherosclerosis and the arterial stiffness of peripheral vessels. The ABI is the ratio of systolic blood pressure measured at the ankle to that measured at the brachial artery in arms by Doppler. ABI was originally proposed for use in the diagnosis of peripheral artery disease (PAD). Although it is a global estimator of whole-limb perfusion, it has also been shown to reflect atherosclerosis in other arterial beds, such that it is an independent marker for coronary artery disease [1]. PWV can be measured between two sites along an arterial tree, such as aortic PWV, carotid–femoral PWV or brachial–ankle PWV (baPWV), with higher values indicating stiffer arteries. The baPWV value may also be used for evaluating large artery damage and peripheral arterial stiffness [2].

Previous studies have reported that the ABI and baPWV potentially reflect cerebrovascular risk in normal subjects. People with low ABI values have a significantly greater risk of developing future stroke than subjects with normal values [3], and baPWV has been shown to be related to silent cerebral infarction and intracranial stenosis [4,5]. A similar risk addition may be observed in stroke patients, even despite the lack of PAD symptoms [6,7,8]. The ABI was reported to be associated with a 1.7- to 2.2-fold higher risk of future stroke, future vascular events, or vascular death in stroke patients [9]. Furthermore, while an ABI cutoff point of 0.9 is commonly used to define abnormal values to identify high-risk individuals, an increased risk of future cardiovascular events was also noted in patients with an ABI value below 1.1 [10]. In addition, an ABI value >1.4 is also a surrogate marker for medial artery calcification and/or PAD [11]. The same is true for the baPWV and the risk of recurrent vascular events in stroke patients. Increased baPWV in patients with acute lacunar infarction has been reported to double the risk of future ischemic stroke [12]. The abovementioned evidence supports the notion that an abnormal ABI or baPWV may be a useful marker to independently predict the cerebrovascular risk in both healthy populations and stroke patients.

Since abnormal ABI and baPWV have been associated with the future risk of cerebrovascular events in normal or stroke subjects, the values of ABI/baPWV may potentially reflect the health status of the cerebral artery. However, it is uncertain whether the ABI and baPWV are associated with cerebral large artery atherosclerosis (LAA) or small vessel disease (SVD) since the involved artery size is different in these two conditions. Controversy exists regarding the associations between the two tests and the two cerebral vascular pathologies [4,5,13,14,15,16,17] in different disease states. In general, it seems that baPWV is related to SVD [4,5,13,14,15,16,17], while the ABI has a stronger correlation with intracranial artery stenosis and extracranial artery stenosis [13,15]. However, the above concepts came from studies performed in healthy subjects. Stroke patients have higher proportions of vascular comorbidities and possible multivessel disease. The crosstalk between PAD and cerebral vascular abnormalities may further redirect the potential interaction of systemic and cerebral atherosclerosis. This study aimed to investigate the associations between ABI and baPWV and the markers of cerebral LAA and SVD in patients with acute ischemic stroke and to extend the above concepts from healthy subjects to stroke patients.

## 2. Materials and Methods

### 2.1. Patients and Image Analysis

This cohort study was conducted at National Cheng Kung University Hospital (NCKUH), a tertiary medical center located in southern Taiwan. A total of 800 to 900 acute ischemic stroke patients are admitted to the NCKU stroke unit annually. Consecutive patients aged more than 20 years who presented to our hospital within 10 days after ischemic stroke onset and had confirmation of the index stroke via new hypodense lesions shown on brain computed tomography (CT) or hyperintense lesions shown on diffusion-weighted magnetic resonance imaging (DWI) were prospectively enrolled from July 2016 to December 2017. Patients who lacked carotid duplex and magnetic resonance imaging (MRI) studies were excluded (Appendix A). Data on patient characteristics, comprising demographic data, previous medical history and comorbidities, stroke severity, treatment strategies, hospital course, and complications, were collected. A self-reported history, medical records, or the current use of drugs for hypertension, diabetes mellitus (DM), or hyperlipidemia were acquired immediately after admission as the reference for past medical history and comorbidities. Additional measurements, including carotid artery duplex, ABI, baPWV, body mass index (BMI), blood pressure, lipid profile, cardiac function, and diabetic profile, were obtained during hospitalization.

The ABI and/or baPWV were measured during hospitalization. To analyze the relationship between the ABI/baPWV and the severity of LAA and SVD, we excluded patients who did not undergo MRI scans or carotid duplex.

### 2.2. ABI/baPWV Acquisition

We used an automated, validated ABI-form device (VS-1000; Fukuda Denshi Co., Ltd., Tokyo, Japan), which simultaneously measures blood pressure in both the arms and the ankles for the measurement. Patients lay in the supine position for 5 min before the measurement. The procedure was performed once for each patient. The patients underwent this screening test during hospitalization or during the first outpatient follow-up visit after discharge from the hospital. BaPWV was obtained simultaneously by the ABI-form devices. All the records were obtained after the patients’ stroke conditions had been stabilized during the hospitalization time within 10 days or during the first outpatient follow-up visits after discharge if ABI/baPWV measurement could not be performed due to physical or clinical reasons. The lowest ABI measured from bilateral sides was taken as the patient’s minimum ABI value and used for analysis. The highest baPWV measured bilaterally was taken as the patient’s maximum baPWV value. The average ABI/baPWV indicated the average of the ABI/baPWV values measured on bilateral sides. Normal range of ABI/baPWV was defined as ABI values between 0.9 and 1.4 and baPWV values less than 14 m/s.

### 2.3. Brain MRI and MRA Acquisition

Brain MRI (PHILIPS Achieva 1.5T, Best, The Netherlands, PHILIPS Ingenia 3T, Amsterdam, The Netherlands and Signa HDxt 1.5T, New York, NY, USA) was performed by technicians in the radiology department during hospitalization for acute stroke. Stroke was confirmed by the presence of hyperintense lesions on diffusion-weighted imaging (DWI). Intracranial magnetic resonance angiography (MRA) was also performed to evaluate the patency of intracranial vessels while performing MRI.

### 2.4. Carotid Duplex Acquisition

The patients underwent a carotid duplex ultrasonography examination (PHILIPS Affiniti 70G, Andover, MA, USA and PHILIPS IE33, USA) during hospitalization. The percentages of bilateral common carotid artery (CCA) and internal carotid artery (ICA) stenosis and plaque severity were recorded.

### 2.5. SVD Markers

We use the “total cerebral SVD score” (CSVD) to represent the extent and severity of SVD. According to a previous study [18], the total CSVD score comprises four components: (1) lacunes, (2) microbleeds, (3) perivascular spaces, and (4) white matter intensities. One point was given for each category if the feature was present. Lacunes were defined as rounded or ovoid lesions, with diameters >3 mm and <20 mm, in the basal ganglia, internal capsule, centrum semiovale, or brainstem, with cerebrospinal fluid (CSF) signal intensity shown on T2-weighted imaging and fluid-attenuated inversion recovery (FLAIR) images, generally with a hyperintense rim shown on FLAIR and no increased signal on DWI [19]. One point was given for the “lacunes” category if any lacunes were present on brain MRI. Microbleeds were defined as small (<5 mm), homogeneous, round hypointense foci on gradient echo images in the cerebellum, brainstem, basal ganglia, white matter, or cortico-subcortical junction, that were differentiated from vessel flow voids and mineral depositions in the globus pallidi [19]. One point was given for the “microbleeds” category if any microbleeds were present on brain MRI. Perivascular spaces were defined as small (<3 mm) punctate (if perpendicular to the plane of the scan) and linear (if longitudinal) hyperintensities on T2-weighted images in the basal ganglia or centrum semiovale [20]. One point was given for the “perivascular spaces” category if moderate to severe perivascular spaces were present on brain MRI. Periventricular white matter hyperintensities (PWMHs) shown on the FLAIR images were defined as leukoaraiosis. Leukoaraiosis was graded as 0 = absent, 1 = “caps” or a thin lining (<0.5 cm), 2 = smooth “halo” (<1 cm), and 3 = irregular PWMH extending into the deep white matter (>1 cm) in accordance with the Fazekas scale [21]. One point was given for the “white matter intensities” category if periventricular Fazekas scale 3 and/or deep white matter Fazekas scale 2–3 intensities were present. Total SVD scores were determined by summing the scores of the aforementioned four categories. The presence of SVD was defined as a total CSVD score higher than 1 (≥1). The severity of SVD was classified as follows: mild, CSVD score = 0; moderate, CSVD score = 1–2; and severe, CSVD score = 3–4.

### 2.6. LAA Markers

The presence of intracranial or extracranial artery stenosis was evaluated by both MRA and carotid duplex ultrasonography. The presence of intracranial or extracranial vessel stenosis on MRA or duplex is graded according to the following criteria: grade 1, normal or mild stenosis (up to 29% diameter stenosis); grade 2, moderate stenosis (30% to 69% diameter stenosis); grade 3, severe stenosis (70% to 100% diameter stenosis); and grade 4, occlusion (100% diameter stenosis with rarefaction of the middle cerebral artery (MCA) distal to the stenosis) [22]. The grading of vessel stenosis was evaluated in the proximal MCAs, proximal anterior cerebral arteries (ACAs), proximal posterior cerebral arteries (PCAs), internal carotid arteries at the siphon, basilar artery, extracranial internal carotid arteries, and common carotid arteries. The stenosis grade was determined (grade 1 = 1.0, grade 2 = 2.0, grade 3 = 3.0, and grade 4 = 4.0). For the extracranial vessels, the stenosis grade was obtained by the diameter ratio of atherosclerotic plaque to inner layer of vessels through carotid duplex. For the intracranial vessels, the stenosis grade was obtained by visual inspection and direct measurement of the diameter ratio of the stenotic part to the nonstenotic proximal parts of the same vessel through MRA. Each vessel’s score was summed to calculate as the total score to reflect the severity of stenosis for subsequent analyses, including intracranial vessels alone, extracranial vessels alone, and the combination of intracranial and extracranial vessels. The lowest possible score for the extracranial vessels was 4 because the scores of the bilateral internal carotid arteries and common carotid arteries were summed. The presence of any extracranial stenosis was defined as any stenosis in the four vessels. Likewise, the lowest possible intracranial vessel stenosis grade was 9 due to the scores of the bilateral ACAs, MCAs, internal carotid arteries (siphon part), PCAs, and basilar artery being summated. The presence of any intracranial artery stenosis was defined as grading scores higher than 9 (>9). The definitions of the severity of extracranial and intracranial vessel stenosis were as follows: mild, each measured vessel stenosis level was <30%; moderate, any measured vessel stenosis level was between 30–69%; and severe, any measured vessel stenosis level was ≥70%.

The imaging data for the markers of cerebral LAA and SVD in the enrolled patients were reviewed by two independent raters (Y.M.C. and T.L.L.). Interrater reliability was obtained by the kappa value prior to the start of the study (kappa value = 0.74) to confirm good interrater agreement.

### 2.7. Statistical Analysis

For demographics and clinical information, the categorical variables are presented as numbers (percentages). Normally distributed quantitative data are presented as the means ± standard deviations (SDs) and nonnormally distributed data are expressed as medians and interquartile ranges (IQRs). To compare proportions across the groups, the Mann–Whitney U test was used due to the nonparametric distribution of our data. Correlation coefficients were calculated by Pearson correlation analysis to evaluate the correlation between the ABI and baPWV and the stenosis grade of extracranial stenosis, intracranial vessels, or total SVD scores. In the multinomial logistic regression analysis, those patients with all intracranial/extracranial vessels defined as showing mild stenosis were classified into the mild vessel stenosis group, while those patients with any intracranial/extracranial vessels defined as showing moderate or severe vessel stenosis were classified into the moderate/severe intracranial/extracranial vessel stenosis group. Then, logistic regression was performed to determine the predictive potential of the ABI and baPWV for the severity of large artery stenosis while using mild vessel stenosis as the reference group. The potential factors adjusted in the logistic regression analysis included age, sex, body mass index, hypertension, diabetes mellitus (DM), hyperlipidemia, and atrial fibrillation. All analyses were performed using the statistical software package SAS version 9.4 (SAS Institute Inc., Cary, NC, USA). A *p* value < 0.05 indicated statistical significance.

## 3. Results

A total of 956 patients were enrolled, and 136 patients were excluded due to insufficient vessel images, either lacking brain MRI, MRA, or carotid duplex (Appendix A). The baseline characteristics of the final 820 patients included in the final analysis are presented in Appendix A. The median age of the included subjects was 68 years old, and they were predominantly male (61.1%). The median average ABI was 1.08, and the baPWV was 18.90. The presence of any extracranial vessel stenosis, intracranial vessel stenosis, or SVD accounted for 90.6%, 67.2%, and 60% of the patients, respectively.

Regarding the relationships between the ABI/baPWV and overall severity of LAA and SVD, Figure 1 demonstrated that extracranial vessel stenosis grades were inversely correlated with the average ABI values (*p* < 0.001) and positively correlated with average baPWV values (*p* < 0.001), indicating that lower ABI and higher baPWV were associated with higher overall extracranial artery stenosis severity. Figure 2 showed that the intracranial vessel stenosis grades were negatively correlated with the average ABI values (*p* < 0.001) and positively correlated with the average baPWV (*p* = 0.004). A similar trend in the correlation between the ABI or baPWV and CSVD score was observed (Figure 3). Regarding the relationship between the ABI or baPWV and CSVD classification, each of the sub-items showed a correlation with ABI or baPWV values, except for the association between the presence of a microbleed and the average ABI (Appendix A).

By using patients with mild extracranial and intracranial artery stenosis as the reference groups, multinomial logistic regression showed that abnormal ABI values independently predicted the presence of moderate (adjusted odds ratio (aOR): 2.18, 95% CI: 1.31–3.63) and severe (aOR: 5.59, 95% CI: 2.21–14.13) extracranial stenosis and severe intracranial stenosis (aOR: 1.89, 95% CI: 1.15–3.11) (Table 1 and Table 2). Older age also independently predicted the presence of moderate and severe extracranial artery stenosis. However, higher baPWV did not predict the presence of severe extracranial or intracranial artery stenosis. Regarding the independent predictors of moderate to severe CSVD, age was the only factor that was significantly associated with moderate to severe CSVD. Neither abnormal ABI nor baPWV predicted the presence of moderate to severe CSVD (Table 3).

## 4. Discussion

In the present study, we noted that an abnormal ABI value independently predicted higher severity of extracranial and intracranial large vessel stenosis. The predictivity of baPWV was less consistent; although baPWV had a positive linear correlation with extracranial and intracranial stenosis, its significance in the associations disappeared when adjustments were made for other risk factors. Although the ABI value and baPWV were correlated with all CSVD subitem scores except for the microbleeds score, the strength of the association between the ABI/baPWV and the severity of CSVD was diminished after adjusting for confounding factors.

In acute stroke patients, ABI measurements are not influenced by stroke-related blood pressure alterations because this measure is a ratio. These measurements are also not influenced by volume changes in relation to intravenous hydration in stroke treatment. The same is true for the measurement of baPWV. However, ABI values are influenced by age, height, and ethnicity [23]. Lower ABI values are correlated with multiple underlying conditions, including old age, hypertension, DM, dyslipidemia, and current smoking [24]. These factors also contribute to the formation of atherosclerosis in cerebral large vessels [25]. Therefore, except for the markers for atherosclerosis in the peripheral artery and coronary artery, the ABI value potentially serves as a marker representing the extent of cerebral artery stenosis. Our study supported this hypothesis and was also in line with previous studies [13,15]. Although a previous study showed the correlation between the ABI value and intracranial artery stenosis was not definitely statistically significant [15], this may be related to limited sample size. After increasing the sample size, we confirmed that the ABI value was also inversely and independently correlated with intracranial artery stenosis in acute stroke patients. This may indicate the universal application of the ABI in predicting the presence of severe extracranial and intracranial cerebral artery vessel stenosis.

Interestingly, controversial results exist regarding the association between baPWV and cerebral vessel conditions. We observed a positive correlation of baPWV with extracranial stenosis grades, but the significance of the correlation disappeared after multivariable adjustment. A lack of consistent correlation between baPWV and intracranial artery stenosis was also noted. A similar phenomenon has been reported in which baPWV was associated with intracranial arterial stenosis in univariate analyses, but the significance vanished after adjustment [4]. These findings may indicate that baPWV may not be a good and universal predictor for cerebral vessel stenosis in stroke patients. The association between baPWV and cerebral large vessel stenosis may be strongly confounded by other conventional stroke risk factors, such as old age and DM, as shown in Table 2.

PAD indicated higher risks of stroke in both primary and secondary prevention settings [26]. The ABI values also had a positive correlation with the risk of stroke recurrence [9]. The higher stroke risk may come from LAA, as reflected by the correlation of the ABI and intra/extracranial vessel stenosis. However, the analysis regarding ABI/baPWV and SVD, contributed by arteriopathy of smaller size, showed diverse results. In our study, we found that there were correlations between the ABI/baPWV values and CSVD subitem scores, but the significance of the correlation diminished after adjusting potential confounding factors for CSVD, especially age. This was consistent with previous studies showing that the ABI value may not be related to white matter changes, silent lacunar infarction, or cerebral SVD [8]. As mentioned above, ABI reflects systemic atherosclerosis and is associated with atherosclerotic risk factors. Although cerebral SVD also results from atherosclerosis involving the cerebral smaller artery, arteriole, venules, and capillaries, also known as arteriosclerosis and cerebral small vascular atherosclerosis [27], the lack of association between ABI and SVD in our study may indicate that the severity of SVD is not necessarily positively correlated with systemic atherosclerosis or cerebral large artery stenosis. Different disease mechanisms or genetic factors in each individual may change the burden of vascular risk factors on vessels of different sizes and vessel territories to atherosclerotic change [28,29,30,31].

Another interesting finding in our study was the lack of association between baPWV and SVD after adjustment. Most previous studies have shown that baPWV is related to SVD and that increased PWV is associated with the presence of white matter lesions and silent lacunar infarction [5,8,12,13,14,15]. However, our results may further suggest that the correlation between baPWV and SVD may be confounded by age factor, which contributed to both an increase in baPWV and the severity of SVD. In addition, the characteristics of the enrolled study population may potentially influence the results. All of our patients had been diagnosed with stroke or TIA. The average baPWV was 18.90 m/s, which was higher than the average baPWV in previous studies including a normal population or a population without clinical stroke (range from 14.2–17.11 m/s) [4,16,17], and similar to the baPWV values in patients with SVD or white matter changes (range from 18.72–20.57 m/s) [12,14,15]. Thus, baPWV may not be a good indicator to predict vessels outside the periphery in patients with relatively higher baPWV.

There are several limitations in this study. First, we excluded 136 patients without sufficient images. Some of these patients could not undergo imaging due to their poor conditions. They may have a higher vascular burden and worse vessel conditions, which may lead to an underestimation of the correlation between ABI/baPWV and cerebral vessel conditions. Second, all of our patients initially presented with acute stroke. Blood pressure and the status of autonomic balance may be altered in the stage of acute stroke compared to the baseline healthy conditions. Although higher BP may not change the ABI values, there are no data regarding changes in ABI values in the acute or chronic stage of stroke. Therefore, our study results reflected the association between ABI/baPWV and cerebral vessel conditions in the acute stage of stroke. Further generalizability from the acute to chronic stage of stroke warrants further investigation. Third, we only measured ABI/baPWV once after admission. Although previous studies showed that within-subject ABI variability may be subtle and could be considered less concerning in comparison with that between different subjects [32,33], multiple measurements could still guarantee better reliability. Fourth, we did not include the period from stroke onset to ABI/baPWV measurement in multivariate analysis due to lack of the exact recording time, but the ABI/baPWV data were all obtained after stroke symptoms stabilized during the hospitalization time within 10 days or during the first outpatient follow-up visit after discharge. Finally, the ABI may underestimate the presence of peripheral vessel atherosclerosis since the measurement may be falsely high. It is uncertain whether such underestimation may occur in the association of ABI values and atherosclerosis in other vascular beds. The toe-brachial index, which provides a more accurate and sensitive risk evaluation than the ABI, especially in patients with diabetes, because it avoids underestimation of medial calcification of peripheral arteries [34]. The toe–brachial index may predict subtle carotid atherosclerosis [35] and also predict the risk of restenosis after angioplasty and stenting procedures [36]. Future studies to evaluate the association of the toe–brachial index and cerebral vessel diseases may be considered.

## 5. Conclusions

In conclusion, our study results reflected the association between ABI/baPWV and cerebral vessel conditions in the acute stage of stroke. We found that an abnormal ABI independently reflects the extent of cerebral large artery stenosis, including extracranial and intracranial vessels, in stroke patients. The association between baPWV and cerebral large artery stenosis was not consistent. The ABI and baPWV did not significantly reflect the severity of CSVD. In clinical practice for stroke care, ABI may serve as a screening tool and warning signal for the existence of extracranial or intracranial atherosclerotic disease.

## Figures and Tables

**Figure 1 diagnostics-13-01455-f001:**
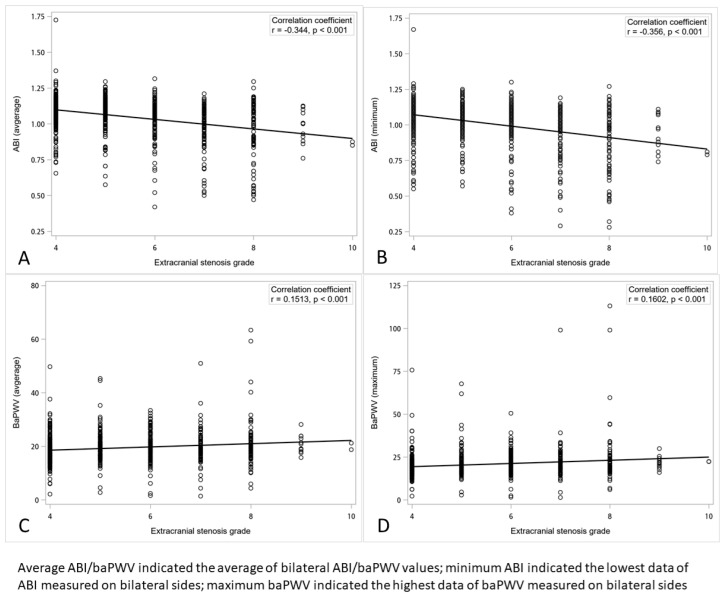
Relationship between the ABI/baPWV and extracranial vessel stenosis. (**A**,**B**) The correlation between the average and minimum ABI and extracranial vessel stenosis. (**C**,**D**) The correlation between the average and maximum baPWV and extracranial vessel stenosis.

**Figure 2 diagnostics-13-01455-f002:**
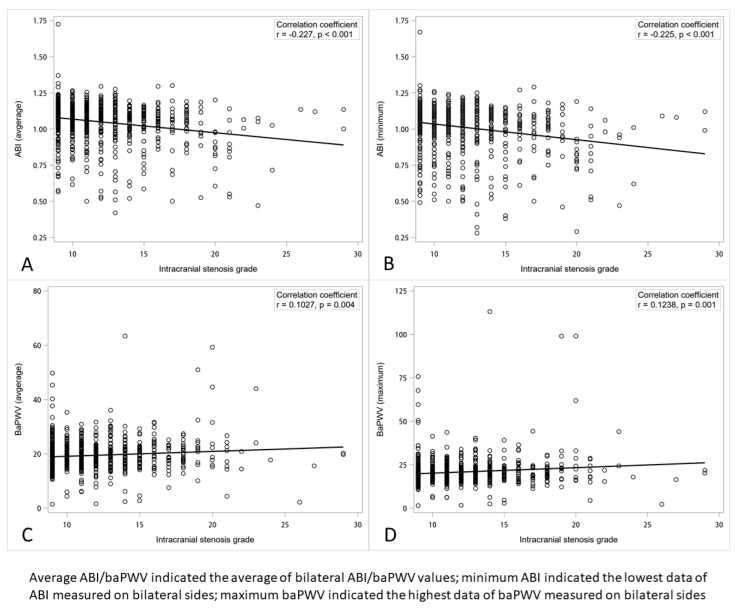
Relationship between the ABI/baPWV and intracranial vessel stenosis. (**A**,**B**) The correlation between the average and minimum ABI and intracranial vessel stenosis. (**C**,**D**) The correlation between the average and maximum baPWV and intracranial vessel stenosis.

**Figure 3 diagnostics-13-01455-f003:**
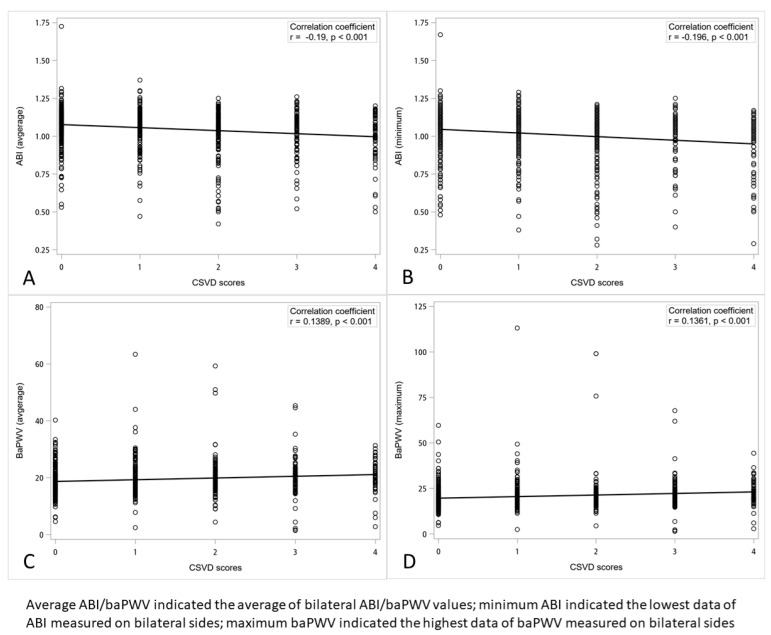
Relationship between the ABI/baPWV and CSVD scores. (**A**,**B**) The correlation between the average and minimum ABI and total CSVD scores. (**C**,**D**) The correlation between the average and maximum baPWV and total CSVD scores.

**Table 1 diagnostics-13-01455-t001:** Multinomial logistic regression for predictors of extracranial vessel stenosis.

	Extracranial Vessel Stenosis Severity											
	Moderate vs. Mild						Severe vs. Mild					
	Crude OR (95% CI)	*p*-Value	Adjusted OR ^a^ (95% CI)	*p*-Value	Adjusted OR ^b^ (95% CI)	*p*-Value	Crude OR (95% CI)	*p*-Value	Adjusted OR ^a^ (95% CI)	*p*-Value	Adjusted OR ^b^ (95% CI)	*p*-Value
ABI: <0.9 or >1.4 vs. 0.9~1.4	3.14 (1.94–5.07)	<0.001	2.10 (1.15–3.83)	0.015	2.18 (1.31–3.63)	0.003	8.52 (3.54–20.49)	<0.001	4.92 (1.60–15.12)	0.005	5.59 (2.21–14.13)	<0.001
Age	1.06 (1.04–1.07)	<0.001	1.07 (1.05–1.09)	<0.001	1.05 (1.04–1.06)	<0.001	1.07 (1.04–1.11)	<0.001	1.07 (1.03–1.12)	0.002	1.06 (1.02–1.10)	0.001
Female vs. Male	1.02 (0.76–1.36)	0.906	0.79 (0.55–1.15)	0.219			0.66 (0.28–1.56)	0.342	0.65 (0.23–1.86)	0.424		
BMI	0.98 (0.94–1.01)	0.150	1.01 (0.97–1.06)	0.655			1.02 (1.00–1.05)	0.107	1.05 (0.99–1.11)	0.127		
HTN	1.00 (0.74–1.36)	0.977	0.63 (0.43–0.94)	0.025			0.91 (0.40–2.05)	0.810	0.37 (0.13–1.05)	0.061		
DM	1.47 (1.09–1.98)	0.012	1.11 (0.75–1.64)	0.607	1.36 (0.99–1.87)	0.054	1.88 (0.85–4.14)	0.119	1.55 (0.56–4.33)	0.403	1.70 (0.76–3.82)	0.197
HL	1.32 (0.99–1.76)	0.057	1.63 (1.14–2.35)	0.008			1.41 (0.64–3.10)	0.400	3.16 (1.14–8.78)	0.027		
Af	0.67 (0.43–1.06)	0.087	0.37 (0.22–0.62)	<0.001			1.22 (0.39–3.84)	0.734	0.74 (0.22–2.50)	0.626		
BaPWV: ≥14 vs. <14	2.68 (1.52–4.74)	0.001	1.57 (0.76–3.25)	0.219	1.52 (0.79–2.90)	0.210	3.30 (0.43–25.07)	0.249	1.48 (0.18–12.21)	0.717	1.73 (0.22–13.73)	0.604
Age	1.06 (1.04–1.07)	<0.001	1.07 (1.05–1.09)	<0.001	1.05 (1.04–1.07)	<0.001	1.07 (1.04–1.11)	<0.001	1.09 (1.04–1.13)	<0.001	1.07 (1.04–1.11)	<0.001
Female vs. Male	1.02 (0.76–1.36)	0.906	0.81 (0.56–1.19)	0.287			0.66 (0.28–1.56)	0.342	0.64 (0.22–1.81)	0.396		
BMI	0.98 (0.94–1.01)	0.150	1.00 (0.96–1.05)	0.933			1.02 (1.00–1.05)	0.107	1.04 (0.98–1.09)	0.191		
HTN	1.00 (0.74–1.36)	0.977	0.64 (0.43–0.97)	0.037			0.91 (0.40–2.05)	0.810	0.34 (0.12–0.94)	0.038		
DM	1.47 (1.09–1.98)	0.012	1.15 (0.77–1.72)	0.483	1.38 (1.00–1.91)	0.050	1.88 (0.85–4.14)	0.119	1.66 (0.60–4.64)	0.331	1.77 (0.79–3.95)	0.167
HL	1.32 (0.99–1.76)	0.057	1.57 (1.08–2.28)	0.018			1.41 (0.64–3.10)	0.400	3.00 (1.09–8.26)	0.034		
Af	0.67 (0.43–1.06)	0.087	0.38 (0.22–0.65)	<0.001			1.22 (0.39–3.84)	0.734	0.80 (0.24–2.67)	0.711		

^a^ Full model. ^b^ Multivariable logistic regression analysis of variables (*p*  <  0.05 in univariate logistic regression analysis). We used minimum ABI/maximum baPWV for analysis in this model. ABI: ankle–brachial index; BMI: body mass index; BaPWV: brachial–ankle pulse wave velocity; HTN: hypertension; DM: diabetes mellitus; HL: hyperlipidemia; Af: atrial fibrillation; OR: odds ratio.

**Table 2 diagnostics-13-01455-t002:** Multinomial logistic regression for predictors of intracranial vessel stenosis.

	Intracranial Stenosis Severity											
	Moderate vs. Mild						Severe vs. Mild					
	Crude OR (95% CI)	*p*-Value	Adjusted OR ^a^ (95% CI)	*p*-Value	Adjusted OR ^b^ (95% CI)	*p*-Value	Crude OR (95% CI)	*p*-Value	Adjusted OR ^a^ (95% CI)	*p*-Value	Adjusted OR ^b^ (95% CI)	*p*-Value
ABI: <0.9 or >1.4 vs. 0.9~1.4	1.65 (0.97–2.81)	0.067	0.94 (0.49–1.78)	0.840	1.25 (0.72–2.18)	0.426	2.47 (1.53–3.99)	<0.001	1.42 (0.80–2.53)	0.229	1.89 (1.15–3.11)	0.012
Age	1.03 (1.02–1.05)	<0.001	1.03 (1.02–1.05)	<0.001	1.03 (1.02–1.05)	<0.001	1.04 (1.02–1.05)	<0.001	1.04 (1.02–1.05)	<0.001	1.03 (1.02–1.05)	<0.001
Female vs. Male	1.25 (0.87–1.79)	0.236	0.95 (0.62–1.48)	0.831			1.31 (0.94–1.84)	0.110	1.19 (0.79–1.78)	0.407		
BMI	0.97 (0.93–1.01)	0.108	0.97 (0.93–1.02)	0.184			0.99 (0.96–1.01)	0.306	0.99 (0.97–1.02)	0.453		
HTN	1.13 (0.80–1.61)	0.483	1.10 (0.69–1.73)	0.697			1.20 (0.86–1.66)	0.279	0.92 (0.60–1.41)	0.712		
DM	1.11 (0.76–1.62)	0.585	1.24 (0.79–1.96)	0.348			1.22 (0.86–1.72)	0.267	1.21 (0.79–1.86)	0.387		
HL	0.80 (0.51–1.27)	0.349	0.91 (0.60–1.38)	0.646			0.74 (0.48–1.13)	0.164	0.95 (0.64–1.41)	0.798		
Af	0.71 (0.40–1.26)	0.244	0.57 (0.30–1.08)	0.085			1.33 (0.83–2.13)	0.241	0.95 (0.55–1.65)	0.862		
BaPWV: ≥14 vs. <14	3.51 (1.57–7.82)	0.002	2.49 (1.08–5.76)	0.033	2.49 (1.09–5.71)	0.031	2.19 (1.18–4.06)	0.013	1.45 (0.74–2.83)	0.278	1.52 (0.79–2.93)	0.213
Age	1.03 (1.02–1.05)	<0.001	1.03 (1.02–1.05)	<0.001	1.03 (1.02–1.05)	<0.001	1.04 (1.02–1.05)	<0.001	1.03 (1.02–1.05)	<0.001	1.04 (1.02–1.05)	<0.001
Female vs. Male	1.25 (0.87–1.79)	0.236	1.08 (0.73–1.58)	0.710			1.31 (0.94–1.84)	0.110	1.12 (0.79–1.61)	0.520		
BMI	0.97 (0.93–1.01)	0.108	0.98 (0.94–1.02)	0.268			0.99 (0.96–1.01)	0.306	0.99 (0.97–1.01)	0.469		
HTN	1.13 (0.80–1.61)	0.483	0.99 (0.67–1.47)	0.968			1.20 (0.86–1.66)	0.279	1.05 (0.72–1.51)	0.815		
DM	1.11 (0.76–1.62)	0.585	1.00 (0.67–1.51)	0.985			1.22 (0.86–1.72)	0.267	1.20 (0.82–1.75)	0.349		
HL	0.80 (0.51–1.27)	0.349	0.79 (0.49–1.29)	0.342			0.74 (0.48–1.13)	0.164	0.70 (0.45–1.11)	0.128		
Af	0.71 (0.40–1.26)	0.244	0.54 (0.30–1.00)	0.050			1.33 (0.83–2.13)	0.241	0.89 (0.53–1.48)	0.644		

^a^ Full model. ^b^ Multivariable logistic regression analysis of variables (*p*  <  0.05 in univariate logistic regression analysis). We used minimum ABI/maximum baPWV for analysis in this model. ABI: ankle–brachial index; BMI: body mass index; BaPWV: brachial–ankle pulse wave velocity; HTN: hypertension; DM: diabetes mellitus; HL: hyperlipidemia; Af: atrial fibrillation; OR: odds ratio.

**Table 3 diagnostics-13-01455-t003:** Multinomial logistic regression for predictors of CSVD.

	CSVD											
	CSVD 1–2 vs. CSVD 0						CSVD 3–4 vs. CSVD 0					
	Crude OR (95% CI)	*p*-Value	Adjusted OR ^a^ (95% CI)	*p*-Value	Adjusted OR ^b^ (95% CI)	*p*-Value	Crude OR (95% CI)	*p*-Value	Adjusted OR ^a^ (95% CI)	*p*-Value	Adjusted OR ^b^ (95% CI)	*p*-Value
ABI: <0.9 or >1.4 vs. 0.9~1.4	2.06 (1.31–3.24)	0.002	1.76 (0.99–3.12)	0.053	1.41 (0.86–2.29)	0.171	2.65 (1.59–4.42)	<0.001	1.66 (0.85–3.23)	0.137	1.42 (0.81–2.49)	0.219
Age	1.05 (1.04–1.07)	<0.001	1.06 (1.04–1.07)	<0.001	1.05 (1.04–1.07)	<<0.001	1.09 (1.07–1.11)	<0.001	1.08 (1.06–1.10)	<0.001	1.08 (1.07–1.10)	<0.001
Female vs. Male	0.83 (0.61–1.14)	0.255	0.74 (0.51–1.10)	0.134			0.84 (0.57–1.24)	0.378	0.64 (0.40–1.04)	0.070		
BMI	1.01 (0.98–1.03)	0.571	1.04 (0.99–1.08)	0.151			0.96 (0.92–1.00)	0.074	0.99 (0.93–1.05)	0.739		
HTN	1.06 (0.78–1.44)	0.695	0.81 (0.54–1.22)	0.320			1.42 (0.98–2.07)	0.067	0.96 (0.58–1.60)	0.877		
DM	1.00 (0.72–1.38)	0.978	1.13 (0.76–1.69)	0.538			0.92 (0.61–1.37)	0.669	1.11 (0.68–1.82)	0.681		
HL	0.79 (0.52–1.19)	0.264	0.76 (0.52–1.10)	0.149			1.15 (0.72–1.84)	0.558	0.73 (0.46–1.16)	0.176		
Af	1.48 (0.94–2.33)	0.088	1.16 (0.68–1.97)	0.587			0.78 (0.41–1.46)	0.437	0.51 (0.24–1.07)	0.073		
BaPWV: ≥14 vs. <14	2.77 (1.43–5.34)	0.002	1.73 (0.84–3.59)	0.139	1.69 (0.83–3.46)	0.151	2.62 (1.14–6.04)	0.024	1.48 (0.57–3.82)	0.419	1.51 (0.59–3.87)	0.389
Age	1.05 (1.04–1.07)	<0.001	1.06 (1.04–1.07)	<0.001	1.05 (1.04–1.07)	<0.001	1.09 (1.07–1.11)	<0.001	1.09 (1.07–1.12)	<0.001	1.09 (1.07–1.11)	<0.001
Female vs. Male	0.83 (0.61–1.14)	0.255	0.65 (0.46–0.93)	0.017			0.84 (0.57–1.24)	0.378	0.61 (0.39–0.94)	0.026		
BMI	1.01 (0.98–1.03)	0.571	1.03 (0.99–1.07)	0.187			0.96 (0.92–1.00)	0.074	1.00 (0.95–1.06)	0.930		
HTN	1.06 (0.78–1.44)	0.695	0.97 (0.68–1.39)	0.868			1.42 (0.98–2.07)	0.067	1.23 (0.79–1.92)	0.366		
DM	1.00 (0.72–1.38)	0.978	0.90 (0.62–1.29)	0.552			0.92 (0.61–1.37)	0.669	0.86 (0.54–1.35)	0.502		
HL	0.79 (0.52–1.19)	0.264	0.91 (0.58–1.43)	0.675			1.15 (0.72–1.84)	0.558	1.29 (0.76–2.20)	0.349		
Af	1.48 (0.94–2.33)	0.088	0.96 (0.58–1.59)	0.867			0.78 (0.41–1.46)	0.437	0.47 (0.24–0.95)	0.034		

^a^ Full model. ^b^ Multivariable logistic regression analysis of variables (*p*  <  0.05 in univariate logistic regression analysis). We used minimum ABI/maximum baPWV for analysis in this model. ABI: ankle–brachial index; BMI: body mass index; BaPWV: brachial–ankle pulse wave velocity; HTN: hypertension; DM: diabetes mellitus; HL: hyperlipidemia; Af: atrial fibrillation; OR: odds ratio.

## Data Availability

All data generated or analyzed during this study are available under request to the corresponding author.

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
