# Peer review of "The Association between Ankle–Brachial Index/Pulse Wave Velocity and Cerebral Large and Small Vessel Diseases in Stroke Patients"

_diagnostics, 2023, doi:10.3390/diagnostics13081455_

Round 1

Reviewer 1 Report

Manuscript review response: “The Association between Ankle-Brachial Index/Pulse Wave Ve-2 locity and Cerebral Large and Small Vessel Diseases in Stroke 3 Patients”.

 Manuscript ID: diagnostics-2259888

Comments

It is an interesting article that could provide information to find a good predictor for cerebral large and small vessel diseases when its used a good combination of non- invasive modalities to diagnosis.

Line 26-28 You talk about there are not Association between Ankle-Brachial Index (ABI)/Pulse Wave Velocity (baPWV)and Cerebral Large and Small Vessel. I understand that both are two global estimates that could perceive stroke in patients. You say that ABI is better estimates than baPWV for detect this pathology, but Neither are independent predictors of cerebral SVD severity. Is important rewriter this idea.

Line 52-57 would you rewrite this paragraph because some result other research s of and ideas they are repeated.

Line 60-61 ABI and baPWV could be markers for independently predict the cerebrovascular risk but I do not understand how could be increased burden if you said are markers, maybe you need rewrite these concepts.

Line 97 you say, “ABI or baPWV was measured during hospitalization” this is correct, or you want to say “ABI or/and baPWV was measured during hospitalization”. My question is because in your title you write “Association between Ankle-Brachial Index/Pulse Wave Ve-2 locity and Cerebral Large and Small Vessel Disease” I understand that you made a relationchip between ABI and baPWV and cerebral large and small vessel.

Line 101-102 In this paragraph “(VS-1000; Fukuda Denshi Co. 101 Ltd.)” you need add the place and country.

Line 103 the word ABI is repeated and is over understood.

Line 115 this paragraph “is also a surrogate marker for medial artery calcification and/or peripheral artery disease” you need add in introduction.

Line 117 the subtitle says “Brain MRI Acquisition” you need change by “Brain MRI and MRA Acquisition”.

Line 117-122 who or who made the acquisition of MRI and MRA.

Line 118 in this paragraph “(PHILIPS Achieva 1.5T, PHILIPS Ingenia 3T and Signa HDxt 1.5T)” you need add the place and country.

Line 124-125 in this text “(PHILIPS Af-124 finiti 70G and PHILIPS IE33)” you need add place and country.

Line 193 in this paragraph “correlation between the 192 ABI or baPWV and the stenosis grade of extracranial stenosis” the correlation is between ABI and stenosis grade of extracranial stenosis or baPWV and stenosis grade of extracranial stenosis or ABI/baPWV (both) and stenosis grade of extracranial stenosis. This is important because in the tittle I understand that is ABI/baPWV (both) and stenosis grade of extracranial stenosis.

Results.  in the images (tables) you could highlight in black the significance in p value.

Line 326-328 you could refer this paragraph with other studies.

Line 343 could you describe the ceiling effect-like phenomenon?

Line 344-345 what kind of changes could you do for adequate baPWV as a a good indicator to predict vessels outside the periphery in patients with relatively higher baPWV? Could you combinate ABI and baPWV for find a best results.

Conclusions: you must be based on the main objective of the study. “our study results reflected the association between ABI/baPWV and cerebral vessel condition in the acute stage of stroke” this information is relevant in your research, is necessary include in your conclusions nad integrate whit prospects in the future.

Author Response

Dear Reviewer 1:

  Thanks for your helpful comments. We had revised our manuscript according to these comments. Thank you very much.

Reviewer 1:

Manuscript review response: “The Association between Ankle-Brachial Index/Pulse Wave Velocity and Cerebral Large and Small Vessel Diseases in Stroke 3 Patients”. Manuscript ID: diagnostics-2259888

Comments 

It is an interesting article that could provide information to find a good predictor for cerebral large and small vessel diseases when its used a good combination of non- invasive modalities to diagnosis.

  1. Line 26-28 You talk about there are not Association between Ankle-Brachial Index (ABI)/Pulse Wave Velocity (baPWV)and Cerebral Large and Small Vessel. I understand that both are two global estimates that could perceive stroke in patients. You say that ABI is better estimates than baPWV for detect this pathology, but Neither are independent predictors of cerebral SVD severity. Is important rewriter this idea.

Reply: Thanks for reviewer’s comment. We rewrite the idea the address this conclusion more precisely. Thank you very much.

“Conclusion: ABI is better than baPWV in screening for and identifying the existence of cerebral large vessel disease, but neither test is a good predictor of cerebral SVD severity.”

  1. Line 52-57 would you rewrite this paragraph because some result other research s of and ideas they are repeated.

Reply: Thanks for reviewer’s comment. We rewrite this section to address this part more precisely. Thank you very much.

“A similar risk addition may be observed in stroke patients, even despite the lack of PAD symptoms [6-8]. The ABI was reported to be associated with a 1.7- to 2.2-fold higher risk of future stroke, future vascular events or vascular death in stroke patients [9]. Furthermore, while an ABI cutoff point of 0.9 is commonly used to define abnormal values to identify high-risk individuals, an increased risk of future cardiovascular events was also noted in patients with an ABI value below 1.1 [10] . In addition, an ABI value >1.4 is also a surrogate marker for medial artery calcification and/or PAD [11].”

  1. Line 60-61 ABI and baPWV could be markers for independently predict the cerebrovascular risk but I do not understand how could be increased burden if you said are markers, maybe you need rewrite these concepts.

Reply: Thanks for reviewer’s comment. We rewrite this section to address this part more precisely. Thank you very much.

“The abovementioned evidence supports the notion that an abnormal ABI or baPWV may be a useful marker to independently predict the cerebrovascular risk in both healthy populations and stroke patients.”

  1. Line 97 you say, “ABI or baPWV was measured during hospitalization” this is correct, or you want to say “ABI or/and baPWV was measured during hospitalization”. My question is because in your title you write “Association between Ankle-Brachial Index/Pulse Wave Velocity and Cerebral Large and Small Vessel Disease” I understand that you made a relationship between ABI and baPWV and cerebral large and small vessel.

Reply: Thanks for reviewer’s comment. We rewrite this section to address this part more precisely. Thanks very much.

“The ABI and/or baPWV was measured during hospitalization. To analyze the relationship between the ABI/baPWV and the severity of LAA and SVD, we excluded patients who did not undergo MRI scans or carotid duplex.

  1. Line 101-102 In this paragraph “(VS-1000; Fukuda Denshi Co. 101 Ltd.)” you need add the place and country.

Reply: Thanks for reviewer’s comment. We added the place and country in this section. Thank you very much.

  1. Line 103 the word ABI is repeated and is over understood.

Reply: Thanks for reviewer’s comment. We rewrite this section. Thank you very much.

“We used an automated, validated ABI-form device (VS-1000; Fukuda Denshi Co. Ltd. Tokyo, Japan), which simultaneously measures blood pressure in both the arms and the ankles for the measurement. Patients lay in the supine position for 5 minutes before the measurement. The procedure was performed once for each patient. The patients underwent this screening test during hospitalization or during the first outpatient follow-up visit after discharge from the hospital.”

  1. Line 115 this paragraph “is also a surrogate marker for medial artery calcification and/or peripheral artery disease” you need add in introduction. 

Reply: Thanks for reviewer’s comment. We added this part in the introduction and removed this sentence from method section. Thank you very much.

“A similar risk addition may be observed in stroke patients, even despite the lack of PAD symptoms [6-8]. The ABI was reported to be associated with a 1.7- to 2.2-fold higher risk of future stroke, future vascular events or vascular death in stroke patients [9]. Furthermore, while an ABI cutoff point of 0.9 is commonly used to define abnormal values to identify high-risk individuals, an increased risk of future cardiovascular events was also noted in patients with an ABI value below 1.1 [10] . In addition, an ABI value >1.4 is also a surrogate marker for medial artery calcification and/or PAD [11].”

  1. Line 117 the subtitle says “Brain MRI Acquisition” you need change by “Brain MRI and MRA Acquisition”.

Reply: Thanks for reviewer’s comment. We added this part into the subtitles. Thank you very much.

  1. Line 117-122 who or who made the acquisition of MRI and MRA.

Reply: Thanks for reviewer’s comment. We added the information regarding the staffs who made the acquisition of MRI and MRA into this section. Thank you very much.

“Brain MRI (PHILIPS Achieva 1.5T, PHILIPS Ingenia 3T and Signa HDxt 1.5T) was performed by the technicians in department of radiology during hospitalization for acute stroke.”

  1. Line 118 in this paragraph “(PHILIPS Achieva 1.5T, PHILIPS Ingenia 3T and Signa HDxt 1.5T)” you need add the place and country.

Reply: Thanks for reviewer’s comment. We added the information in this section .Thank you very much.

  1. Line 124-125 in this text “(PHILIPS Af-124 finiti 70G and PHILIPS IE33)” you need add place and country.

Reply: Thanks for reviewer’s comment. We added the information in this section .Thank you very much.

  1. Line 193 in this paragraph “correlation between the ABI or baPWV and the stenosis grade of extracranial stenosis” the correlation is between ABI and stenosis grade of extracranial stenosis or baPWV and stenosis grade of extracranial stenosis or ABI/baPWV (both) and stenosis grade of extracranial stenosis. This is important because in the tittle I understand that is ABI/baPWV (both) and stenosis grade of extracranial stenosis.

Reply: Thanks for reviewer’s comment. We changed the word “or” to “and” because both tests and stenosis grade were performed. Thank you very much.

“Correlation coefficients were calculated by Pearson correlation analysis to evaluate the correlation between the ABI and baPWV and the stenosis grade of extracranial stenosis, intracranial vessels or total SVD scores.”

  1. in the images (tables) you could highlight in black the significance in p value.

Reply: Thanks for reviewer’s comment. We highlighted the significant p values in bold and black in the images and tables. Thank you very much.

  1. Line 326-328 you could refer this paragraph with other studies.

Reply: Thanks for reviewer’s comment. We added four references [28-31] to support this viewpoint. Thank you very much.

[28] Danese, C., et al., Do hypertension and diabetes mellitus influence the site of

atherosclerotic plaques? Clin Ter, 2006. 157(1): p. 9-13.

[29] Kayashima, Y. and N. Maeda-Smithies, Atherosclerosis in Different Vascular

Locations Unbiasedly Approached with Mouse Genetics. Genes (Basel), 2020. 11(12).

[30] Garrett, N.E., 3rd, et al., Genetic analysis of a mouse cross implicates an

anti-inflammatory gene in control of atherosclerosis susceptibility. Mamm Genome, 2017. 28(3-4): p. 90-99.

[31] Suzuki, T., et al., Plaque regression in one artery is not necessarily associated with parallel changes in other vascular beds. Heart Vessels, 2011. 26(3): p. 242-51.

“Different disease mechanisms or genetic factors in each individual may change the burden of vascular risk factors on vessels of different sizes and vessel territories to atherosclerotic change [28-31].”

  1. Line 343 could you describe the ceiling effect-like phenomenon? 

Reply: Thanks for reviewer’s comment. According to the distribution of baPWV in our patient population, we noted that our patients had higher average values of baPWV (Our average baPWV was 19.41 m/s.). This value was higher than the average baPWV in previous studies that included a normal population or a population without clinical stroke (range from 14.2-17.11 m/s) and more similar with the studies recruiting patient groups with SVD or white matter change (baPWV range from 18.72-20.57 m/s). Therefore, the lack of association between baPWV and SVD may be confounded by the baseline higher values of baPWV in our study. For better explaining this viewpoint, we rewrite this section for clarification.

“In addition, the characteristics of the enrolled study population may potentially influence the results. All of our patients had been diagnosed with stroke or TIA. The average baPWV was 18.90 m/s, which was higher than the average baPWV in previous studies including a normal population or a population without clinical stroke (range from 14.2-17.11 m/s) [4, 16, 17], and similar to the baPWV values in patients with SVD or white matter changes (range from 18.72-20.57 m/s) [12, 14, 15]. Thus, baPWV may not be a good indicator to predict vessels outside the periphery in patients with relatively higher baPWV. “

  1. Line 344-345 what kind of changes could you do for adequate baPWV as a good indicator to predict vessels outside the periphery in patients with relatively higher baPWV? Could you combinate ABI and baPWV for find a best results.

Reply: Thanks for reviewer’s comment. According to the comment, we combined ABI and baPWV to find the correlation between the single marker or combined markers and cerebrovascular pathology. The data were presented as below:

Table S4-1. Multinomial logistic regression for predictors for extracranial vessel stenosis.

Extracranial vessel stenosis severity

            moderate vs. mild           severe vs. mild

Crude OR

(95% CI)

p-value

Adjusted OR

(95% CI)

p-value

Crude OR

(95% CI)

p-value

Adjusted OR

(95% CI)

p-value

Group

A=0,B=0

Ref.

Ref.

Ref.

Ref.

A=0,B=1

3.61 (1.81-7.21)

<0.001

1.99 (0.93-4.25)

0.077

1.79 (0.23-14.05)

0.580

0.90 (0.11-7.42)

0.924

A=1,B=0

5.44 (1.40-21.11)

0.014

3.83 (0.80-18.33)

0.093

-

-

-

-

A=1,B=1

11.52 (4.91-27.00)

<0.001

4.25 (1.68-10.78)

0.002

18.77 (2.22-158.36)

0.007

5.95 (0.65-54.84)

0.115

Age

1.06 (1.04-1.07)

<0.001

1.05 (1.04-1.06)

<0.001

1.07 (1.04-1.11)

<0.001

1.06 (1.02-1.10)

0.002

Female vs. Male

1.02 (0.76-1.36)

0.906

0.66 (0.28-1.56)

0.342

BMI

0.98 (0.94-1.01)

0.150

1.02 (1.00-1.05)

0.107

HTN

1.00 (0.74-1.36)

0.977

0.91 (0.40-2.05)

0.810

DM

1.47 (1.09-1.98)

0.012

1.34 (0.97-1.86)

0.075

1.88 (0.85-4.14)

0.119

1.66 (0.73-3.76)

0.226

HL

1.32 (0.99-1.76)

0.057

1.41 (0.64-3.10)

0.400

Af

0.67 (0.43-1.06)

0.087

1.22 (0.39-3.84)

0.734

Group interpretation: A=0 (ABI: 0.9~1.4), A=1 (ABI<0.9 or ABI>1.4); B=0 (BaPWV<14), B=1 (BaPWV³14)

Multivariable logistic regression analysis of variables (p < 0.05 in univariate logistic regression analysis).

*We used minimum ABI/maximum baPWV for analysis in this model.

ABI: Ankle-brachial index; BMI: body mass index; baPWV: brachial-ankle pulse wave velocity; HTN: hypertension; DM: diabetes mellitus; HL: hyperlipidemia; Af: atrial fibrillation; OR: odds ratio.

Table S4-2. Multinomial logistic regression for predictors of intracranial vessel stenosis.

Intracranial stenosis severity

moderate vs. mild

severe vs. mild

Crude OR

(95% CI)

p-value

Adjusted OR

(95% CI)

p-value

Crude OR

(95% CI)

p-value

Adjusted OR

(95% CI)

p-value

Group

A=0,B=0

Ref.

Ref.

Ref.

Ref.

A=0,B=1

5.18 (1.77-15.21)

0.003

3.62 (1.21-10.86)

0.022

1.80 (0.91-3.58)

0.092

1.24 (0.61-2.56)

0.553

A=1,B=0

4.00 (0.77-20.81)

0.100

2.84 (0.52-15.62)

0.230

0.86 (0.19-3.98)

0.844

0.60 (0.12-2.96)

0.529

A=1,B=1

10.94 (3.25-36.77)

<0.001

5.98 (1.70-20.98)

0.005

6.15 (2.63-14.40)

<0.001

3.29 (1.33-8.16)

0.010

Age

1.03 (1.02-1.05)

<0.001

1.03 (1.01-1.04)

<0.001

1.04 (1.02-1.05)

<0.001

1.03 (1.02-1.04)

<0.001

Female vs. Male

1.25 (0.87-1.79)

0.236

1.31 (0.94-1.84)

0.110

BMI

0.97 (0.93-1.01)

0.108

0.99 (0.96-1.01)

0.306

HTN

1.13 (0.80-1.61)

0.483

1.20 (0.86-1.66)

0.279

DM

1.11 (0.76-1.62)

0.585

1.22 (0.86-1.72)

0.267

HL

0.80 (0.51-1.27)

0.349

0.74 (0.48-1.13)

0.164

Af

0.71 (0.40-1.26)

0.244

1.33 (0.83-2.13)

0.241

Group interpretation: A=0 (ABI: 0.9~1.4), A=1 (ABI<0.9 or ABI>1.4); B=0 (BaPWV<14), B=1 (BaPWV³14)

Multivariable logistic regression analysis of variables (p < 0.05 in univariate logistic regression analysis).

*We used minimum ABI/maximum baPWV for analysis in this model.

ABI: Ankle-brachial index; BMI: body mass index; baPWV: brachial-ankle pulse wave velocity; HTN: hypertension; DM: diabetes mellitus; HL: hyperlipidemia; Af: atrial fibrillation; OR: odds ratio.

Table S4-3. Multinomial logistic regression for predictors of CSVD.

CSVD

CSVD 1-2 vs. CSVD 0

CSVD 3-4 vs. CSVD 0

Crude OR

(95% CI)

p-value

Adjusted ORb

(95% CI)

p-value

Crude OR

(95% CI)

p-value

Adjusted ORb

(95% CI)

p-value

Group

A=0,B=0

Ref.

Ref.

Ref.

Ref.

A=0,B=1

4.16 (1.80-9.61)

0.001

2.49 (1.03-6.06)

0.043

3.56 (1.23-10.32)

0.019

1.90 (0.59-6.10)

0.283

A=1,B=0

6.64 (1.47-30.00)

0.014

5.29 (0.97-28.93)

0.055

5.81 (0.94-36.00)

0.059

3.47 (0.42-28.67)

0.249

A=1,B=1

8.37 (3.26-21.49)

<0.001

3.37 (1.23-9.25)

0.018

8.90 (2.78-28.44)

<0.001

2.46 (0.68-8.88)

0.168

Age

1.05 (1.04-1.07)

<0.001

1.05 (1.04-1.07)

<0.001

1.09 (1.07-1.11)

<0.001

1.08 (1.06-1.11)

<0.001

Female vs. Male

0.83 (0.61-1.14)

0.255

0.84 (0.57-1.24)

0.378

BMI

1.01 (0.98-1.03)

0.571

0.96 (0.92-1.00)

0.074

HTN

1.06 (0.78-1.44)

0.695

1.42 (0.98-2.07)

0.067

DM

1.00 (0.72-1.38)

0.978

0.92 (0.61-1.37)

0.669

HL

0.79 (0.52-1.19)

0.264

1.15 (0.72-1.84)

0.558

Af

1.48 (0.94-2.33)

0.088

0.78 (0.41-1.46)

0.437

Group interpretation: A=0 (ABI: 0.9~1.4), A=1 (ABI<0.9 or ABI>1.4); B=0 (BaPWV<14), B=1 (BaPWV³14)

Multivariable logistic regression analysis of variables (p < 0.05 in univariate logistic regression analysis).

*We used minimum ABI/maximum baPWV for analysis in this model.

ABI: Ankle-brachial index; BMI: body mass index; baPWV: brachial-ankle pulse wave velocity; HTN: hypertension; DM: diabetes mellitus; HL: hyperlipidemia; Af: atrial fibrillation; OR: odds ratio.

Because of the patient number in each group may be too few to have a stable statistical analysis, wide 95% CI (like A1B1 in the analysis of extracranial artery stenosis) may be noted. In some part, the patient number is too few to be presented (like A1B0 in the analysis of extracranial artery stenosis). Because of unstable statistical analysis, we did not present this data into formal manuscript. Thanks for your helpful comments. 

  1. Conclusions: you must be based on the main objective of the study. “our study results reflected the association between ABI/baPWV and cerebral vessel condition in the acute stage of stroke” this information is relevant in your research, is necessary include in your conclusions and integrate with prospects in the future.

Reply: Thanks for reviewer’s precise comments. We added this part into our conclusion. Thanks very much.

“In conclusion, our study results reflected the association between ABI/baPWV and cerebral vessel conditions in the acute stage of stroke. We found that an abnormal ABI independently reflects the extent of cerebral large artery stenosis, including extracranial and intracranial vessels, in stroke patients. The association between baPWV and cerebral large artery stenosis was not consistent. The ABI and baPWV did not significantly reflect the severity of CSVD. In clinical practice for stroke care, ABI may serve as a screening tool and warning signal for the existence of extracranial or intracranial atherosclerotic disease.”

Reviewer 2 Report

I have received for review an original research article entitled “The Association between Ankle-Brachial Index/Pulse Wave Velocity and Cerebral Large and Small Vessel Diseases in Stroke Patients” prepared by Yu-Ming Chang et al., which is being processed for publication in the journal Diagnostics (IF=3.992). Cardiovascular diseases, including stroke, are one of the most important public health problems worldwide. Therefore, the effort of the Authors involved in addressing such an important topic should be appreciated. The submitted manuscript has quite a high scientific and cognitive value and should be considered for publication in the future. However, some significant improvements are needed. My suggestions are listed below.

1)     The introduction well describes the basic information about the ankle-brachial index and the measurement of pulse wave velocity. However, I think it would be worth discussing the limitations of the ankle-brachial index a bit further, including when it may be falsely high. It is worth mentioning that in such situations, the toe-brachial index is a valuable supplement to the measurement of the ankle-brachial index. It is worth mentioning that the measurement of the toe-brachial index is especially important in people with diabetes. Diabetes is a strong factor that accelerates the development of atherosclerosis and modifies its course. It is also a risk factor for restenosis after angioplasty and stenting procedures. (10.3390/ijerph17249339; 10.3390/ijerph182211970)

2)     As far as the description of the methodology of statistical analysis is concerned, there is no information on how the compliance of quantitative variables with normal distribution was tested.

3)     In my opinion, in the case of variables whose distribution differs significantly from the normal distribution, a more adequate measure of central tendency and dispersion is the median and interquartile range, rather than the arithmetic mean and standard deviation. Please respond to this statement.

4)     The intention of the Authors was that in the range of ABI values, it was examined how an abnormal value (decreased or elevated, treated together) affects the risk of the presence of cerebrovascular pathology in relation to the normal ABI value. I think the results of the analysis would be interesting if we considered additionally the lowered vs. normal ABI and the elevated vs. normal ABI separately. Did the Authors attempt to conduct such an analysis at the stage of material preparation before the preparation of the manuscript?

5)     The list of references should be prepared in accordance with the editorial rules of the MDPI publishing house.

Author Response

Dear Reviewer 2:

    Thanks for your helpful comments. We have revised our manuscript and statistical analysis according to comments. Thank you very much.

Reviewer 2:

I have received for review an original research article entitled “The Association between Ankle-Brachial Index/Pulse Wave Velocity and Cerebral Large and Small Vessel Diseases in Stroke Patients” prepared by Yu-Ming Chang et al., which is being processed for publication in the journal Diagnostics (IF=3.992). Cardiovascular diseases, including stroke, are one of the most important public health problems worldwide. Therefore, the effort of the Authors involved in addressing such an important topic should be appreciated. The submitted manuscript has quite a high scientific and cognitive value and should be considered for publication in the future. However, some significant improvements are needed. My suggestions are listed below.

(1)The introduction well describes the basic information about the ankle-brachial index and the measurement of pulse wave velocity. However, I think it would be worth discussing the limitations of the ankle-brachial index a bit further, including when it may be falsely high. It is worth mentioning that in such situations, the toe-brachial index is a valuable supplement to the measurement of the ankle-brachial index. It is worth mentioning that the measurement of the toe-brachial index is especially important in people with diabetes. Diabetes is a strong factor that accelerates the development of atherosclerosis and modifies its course. It is also a risk factor for restenosis after angioplasty and stenting procedures. (10.3390/ijerph17249339; 10.3390/ijerph182211970)

Reply: Thanks for reviewer’s helpful comment. We added this viewpoint into our limitation section to further discuss the limitation of ABI and the consideration to use toe-brachial index as future study direction. Thanks very much.

“Finally, the ABI may underestimate the presence of peripheral vessel atherosclerosis since the measurement may be falsely high. It is uncertain whether such underestimation may occur in the association of ABI values and atherosclerosis in other vascular beds. The toe-brachial index, which provides a more accurate and sensitive risk evaluation than the ABI, especially in patients with diabetes, because it avoids underestimation of medial calcification of peripheral arteries [34]. The toe-brachial index may predict subtle carotid atherosclerosis [35] and also for the risk of restenosis after angioplasty and stenting procedures [36]. Future studies to evaluate the association of the toe-brachial index and cerebral vessel diseases may be considered.”

(2)As far as the description of the methodology of statistical analysis is concerned, there is no information on how the compliance of quantitative variables with normal distribution was tested.   In my opinion, in the case of variables whose distribution differs significantly from the normal distribution, a more adequate measure of central tendency and dispersion is the median and interquartile range, rather than the arithmetic mean and standard deviation. Please respond to this statement.

Reply: Thanks for reviewer’s helpful comment. We checked the data (quantitative variables, including age, BMI, the distribution of CSVD, ECAD, ICAD stenosis grade, ABI and baPWV) to confirm whether these data are normally distributed. Because we found that these data were nonnormally distributed in this dataset, we then used median (IQR) to present the data in the demographic table as reviewer’s comment. Thanks very much.

3)   The intention of the Authors was that in the range of ABI values, it was examined how an abnormal value (decreased or elevated, treated together) affects the risk of the presence of cerebrovascular pathology in relation to the normal ABI value. I think the results of the analysis would be interesting if we considered additionally the lowered vs. normal ABI and the elevated vs. normal ABI separately. Did the Authors attempt to conduct such an analysis at the stage of material preparation before the preparation of the manuscript?

Reply: Thanks for reviewer’s helpful comments. Before the preparation of the manuscript, we had performed the analysis regarding the association between cerebrovascular pathology and abnormal ABI, including lower than normal ABI (< 0.9 vs. 0.9-1.4) and higher than normal ABI (>1.4 vs. 0.9-1.4). The data were presented in the below section:

Table 1-1. Multinominal logistic regression for predictors for extracranial vessel stenosis.

Extracranial vessel stenosis severity

        moderate vs. mild                  severe vs mild                

Crude OR

(95% CI)

p-value

Adjusted OR

(95% CI)

p-value

Crude OR

(95% CI)

p-value

Adjusted OR

(95% CI)

p-value

ABI: >1.4 vs. 0.9~1.4

0.74 (0.10-5.26)

0.760

0.46 (0.06-3.40)

0.447

-

-

-

-

ABI: <0.9 vs. 0.9~1.4

3.37 (2.05-5.53)

<0.001

2.36 (1.39-4.00)

0.001

9.33 (3.85-22.64)

<0.001

6.17 (2.42-15.76)

<0.001

Age

1.06 (1.04-1.07)

<0.001

1.05 (1.04-1.07)

<0.001

1.07 (1.04-1.11)

<0.001

1.06 (1.02-1.10)

0.001

Female vs. Male

1.02 (0.76-1.36)

0.906

0.66 (0.28-1.56)

0.342

BMI

0.98 (0.94-1.01)

0.150

1.02 (1.00-1.05)

0.107

HTN

1.00 (0.74-1.36)

0.977

0.91 (0.40-2.05)

0.810

DM

1.47 (1.09-1.98)

0.012

1.36 (0.99-1.86)

0.059

1.88 (0.85-4.14)

0.119

1.68 (0.75-3.77)

0.209

HL

1.32 (0.99-1.76)

0.057

1.41 (0.64-3.10)

0.400

Af

0.67 (0.43-1.06)

0.087

1.22 (0.39-3.84)

0.734

Multivariable logistic regression analysis of variables (p < 0.05 in univariate logistic regression analysis).

- indicated the person in this part was too few to be presented

*We used minimum ABI/maximum baPWV for analysis in this model.

ABI: Ankle-brachial index; BMI: body mass index; baPWV: brachial-ankle pulse wave velocity; HTN: hypertension; DM: diabetes mellitus; HL: hyperlipidemia; Af: atrial fibrillation; OR: odds ratio.

Table 2-1. Multinominal logistic regression for predictors of intracranial vessel stenosis.

Intracranial stenosis severity

moderate vs. mild              Severe vs. mild

Crude OR

(95% CI)

p-value

Adjusted OR

(95% CI)

p-value

Crude OR

(95% CI)

p-value

Adjusted OR

(95% CI)

p-value

ABI: >1.4 vs. 0.9~1.4

0.62 (0.06-6.86)

0.695

0.44 (0.04-4.97)

0.508

0.48 (0.04-5.37)

0.555

0.35 (0.03-3.91)

0.392

ABI: <0.9 vs. 0.9~1.4

1.73 (1.00-2.99)

0.0498

1.32 (0.75-2.32)

0.339

2.63 (1.61-4.30)

<0.001

2.02 (1.21-3.36)

0.007

Age

1.03 (1.02-1.05)

<0.001

1.03 (1.02-1.05)

<0.001

1.04 (1.02-1.05)

<0.001

1.03 (1.02-1.05)

<0.001

Female vs. Male

1.25 (0.87-1.79)

0.236

1.31 (0.94-1.84)

0.110

BMI

0.97 (0.93-1.01)

0.108

0.99 (0.96-1.01)

0.306

HTN

1.13 (0.80-1.61)

0.483

1.20 (0.86-1.66)

0.279

DM

1.11 (0.76-1.62)

0.585

1.22 (0.86-1.72)

0.267

HL

0.80 (0.51-1.27)

0.349

0.74 (0.48-1.13)

0.164

Af

0.71 (0.40-1.26)

0.244

1.33 (0.83-2.13)

0.241

Multivariable logistic regression analysis of variables (p < 0.05 in univariate logistic regression analysis).

*We used minimum ABI/maximum baPWV for analysis in this model.

ABI: Ankle-brachial index; BMI: body mass index; baPWV: brachial-ankle pulse wave velocity; HTN: hypertension; DM: diabetes mellitus; HL: hyperlipidemia; Af: atrial fibrillation; OR: odds ratio.

Table 3-1. Multinominal logistic regression for predictors of CSVD.

CSVD

moderate vs. mild              severe vs.mild

Crude OR

(95% CI)

p-value

Adjusted OR

(95% CI)

p-value

Crude OR

(95% CI)

p-value

Adjusted OR

(95% CI)

p-value

ABI: >1.4 vs. 0.9~1.4

0.37 (0.04-3.59)

0.392

0.21 (0.02-2.09)

0.183

-

-

-

-

ABI: <0.9 vs. 0.9~1.4

2.23 (1.39-3.56)

0.001

1.54 (0.93-2.54)

0.091

2.92 (1.73-4.92)

<0.001

1.59 (0.90-2.82)

0.113

Age

1.05 (1.04-1.07)

<0.001

1.05 (1.04-1.07)

<0.001

1.09 (1.07-1.11)

<0.001

1.08 (1.07-1.10)

<0.001

Female vs. Male

0.83 (0.61-1.14)

0.255

0.84 (0.57-1.24)

0.378

BMI

1.01 (0.98-1.03)

0.571

0.96 (0.92-1.00)

0.074

HTN

1.06 (0.78-1.44)

0.695

1.42 (0.98-2.07)

0.067

DM

1.00 (0.72-1.38)

0.978

0.92 (0.61-1.37)

0.669

HL

0.79 (0.52-1.19)

0.264

1.15 (0.72-1.84)

0.558

Af

1.48 (0.94-2.33)

0.088

0.78 (0.41-1.46)

0.437

Multivariable logistic regression analysis of variables (p < 0.05 in univariate logistic regression analysis).

- indicated the person in this part was too few to be presented

*We used minimum ABI/maximum baPWV for analysis in this model.

ABI: Ankle-brachial index; BMI: body mass index; baPWV: brachial-ankle pulse wave velocity; HTN: hypertension; DM: diabetes mellitus; HL: hyperlipidemia; Af: atrial fibrillation; OR: odds ratio.

These data indicated that the association between ABI lower than 0.9 and extra/intracranial artery stenosis was higher than ABI greater than 1.4. However, because ABI >1.4 is also a surrogate marker for medial artery calcification and/or peripheral artery disease and the patient amount of ABI > 1.4 is too few to identify the true association. Therefore, we still decided to define the normal ABI population as patients with ABI between 0.9-1.4 and abnormal ABI group (ABI < 0.9 or ABI > 1.4). Thanks for reviewer’s comments.

4)     The list of references should be prepared in accordance with the editorial rules of the MDPI publishing house.

Reply: Thanks for reviewer’s comment. We rechecked the reference to meet the MPDI rules. Thanks very much.

Round 2

Reviewer 1 Report

I don´t have more comments or suggestions for authors 

Reviewer 2 Report

The paper has been improved. I have no further comments. I recommend it for publication in its current form. Congratulations for Authors and good luck in their further scientific work and clinical practice.